# The Role of TLR2 in Infectious Diseases Caused by Mycobacteria: From Cell Biology to Therapeutic Target

**DOI:** 10.3390/biology11020246

**Published:** 2022-02-05

**Authors:** Wanbin Hu, Herman P. Spaink

**Affiliations:** Institute of Biology Leiden, Animal Science and Health, Leiden University, Einsteinweg 55, 2333 CC Leiden, The Netherlands; w.hu@biology.leidenuniv.nl

**Keywords:** TLR2, TLR2 ligands, leukocyte biology, *Mycobacterium tuberculosis*, nontuberculous mycobacteria, therapeutic target

## Abstract

**Simple Summary:**

Due to the broad functions of Toll-like receptor 2 (TLR2) in innate immunity, the drive for the development of TLR2-targeted therapeutic treatments has accelerated in recent decades. However, its dual role in both the activation and suppression of innate immune responses makes it very difficult to use the results from gathered basic research and apply them to the development of clinical trials. Therefore, this review aims to summarize the knowledge of the function of TLR2 in innate immunity and metabolism to provide some future research directions.

**Abstract:**

Innate immunity is considered the first line of defense against microbial invasion, and its dysregulation can increase the susceptibility of hosts to infections by invading pathogens. Host cells rely on pattern recognition receptors (PRRs) to recognize invading pathogens and initiate protective innate immune responses. Toll-like receptor 2 (TLR2) is believed to be among the most important Toll-like receptors for defense against mycobacterial infection. TLR2 has been reported to have very broad functions in infectious diseases and also in other diseases, such as chronic and acute inflammatory diseases, cancers, and even metabolic disorders. However, TLR2 has an unclear dual role in both the activation and suppression of innate immune responses. Moreover, in some studies, the function of TLR2 was shown to be controversial, and therefore its role in several diseases is still inconclusive. Therefore, although TLR2 has been shown to have an important function in innate immunity, its usefulness as a therapeutic target in clinical application is still uncertain. In this literature review, we summarize the knowledge of the functions of TLR2 in host–mycobacterial interactions, discuss controversial results, and suggest possibilities for future research.

## 1. Introduction

### 1.1. Innate Immunity and Toll-like Receptors

Animal cells rely on germline-encoded pattern recognition receptors (PRRs) to initiate protective innate immune responses [1,2]. PRRs recognize invading microbial pathogens through pathogen-associated molecular patterns (PAMPs) from the pathogens in combination with the recognition of danger-associated molecular patterns (DAMPs) produced by infected or damaged tissues [1,3,4]. PRRs can be divided into the following 8 well-characterized groups: Toll-like receptors (TLRs), retinoic acid-inducible gene-I (RIG)-I-like receptors (RLRs), nucleotide-binding oligomerization domain (NOD)-like receptors (NLRs), C-type lectin receptors (CLRs), opsonic receptors, AIM2-like receptors (ALRs), scavenger receptors (SRs), and stimulator of interferon genes (STING) [5,6,7]. TLRs are the most widely studied PRRs, as shown in Figure 1.

### 1.2. The Diversity of TLRs

The function of TLRs has been studied extensively in recent decades [8] (Figure 1). Their capacity as key initiators of innate immune responses makes them attractive therapeutic targets [9,10,11]. TLRs are homologs of the Toll gene that was first discovered to be involved in embryonic development in *Drosophila* [12,13]. The investigation of TLRs intensified after their function in defense against microbial infection in *Drosophila* and vertebrates was demonstrated [14]. TLRs are characterized as type Ⅰ transmembrane proteins, consisting of an outside membrane N-terminal ectodomain, a single transmembrane domain, and a C-terminal domain inside the membrane [15,16,17]. The N-terminal ectodomain contains leucine-rich repeats (LRRs) and selectively recognizes PAMPs and DAMPs, while the C-terminal domain, also known as a cytoplasmic domain, comprises an evolutionary conserved Toll/IL-1 receptor (TIR) homology domain that is responsible for signal transduction [18,19,20]. We summarize the different PAMPs and DAMPs that are recognized by specific TLRs in Figure 2.

Different animal species have different numbers of genes that encode TLRs. For instance, in the human genome, 10 TLRs are encoded, whereas the mouse and zebrafish genomes encode at least 12 and 20 TLRs, respectively [4,17,21,22]. In teleost fish, some TLRs have diversified to recognize the same category of PAMPs, for instance in the case of recognition of CpGs by TLR9 and TLR21 [23]. TLRs can be divided into two subgroups based on their cellular location. TLRs are expressed either on the cell surface or in intracellular compartments. In humans, TLR1, 2, 4, 5, 6, and 10 are expressed on the cell surface, while TLR3, 7, 8, and 9 are localized in intracellular membranes [22]. In mice, the cellular distribution of the conserved TLRs is assumed to be the same as the distribution in humans. The mouse -specific TLR12 is an endolysosomal TLR and has a function in responding to profilin from pathogens by cooperating with TLR11 [24,25]. In addition, it has been demonstrated that TLR11 recognizes flagellin by using a different domain to that used for profilin recognition [26]. However, the susceptibility of mouse TLR11 mutants to *Salmonella typhi* infection is still controversial [27,28]. TLR13 is another additional TLR in mice that is expressed in intracellular compartments, and has been reported to bind RNA from *Staphylococcus* and *Streptococcus* spp. [29].

Crystal structures of TLR-ligand complexes have been widely studied [30]. It has been shown that oligomerization states can be different between TLR receptors that recognize the same ligand. For instance, this is the case with TLR3 and TLR7 binding the signaling regulator chaperone Unc-93 homolog B1 [31]. In addition, the orthologs of TLRs in different species have differences in structures that lead to functional diversity [32,33,34]. For instance, the structure of TLR2 in humans and mice is different in its ligand binding domains, which makes some mouse TLR2 agonists not effective in humans [33]. At present, there are still challenges to design TLR2-selective agonists or antagonists that are active in humans based on structures of the TLR2-ligand complexes and alternative animal testing models [35,36]. 

### 1.3. TLR2, an Important Member of the TLR Family

After the identification of TLR2 in 1998, much progress has been made in our understanding of its function [9,38,39,46]. Its functions in the recognition of a large number of ligands, including PAMPs and DAMPs, are complicating the studies of the underlying mechanisms of recognition. In addition, the widely distribution of TLR2 on various types of cells, e.g., immune, endothelial, and epithelial cells, also underlines its wide range of functions [39]. Considering the broad functions of TLR2, the drive for the development of TLR2-related therapeutic targeted vaccine or treatment has accelerated in recent decades [9,46,47]. However, some studies on the role of TLR2 in infectious diseases are still controversial [9]. Moreover, its function in immune regulation in the other diseases is still poorly understood [48]. For example, TLR2 plays a dual role in infection processes [48,49]. TLR2 has been shown to play a protective role during infection by triggering a strong pro-inflammatory response, which is considered as beneficial for bacterial clearance [50,51]. However, the excessive inflammation caused by TLR2 can lead to tissue damage and even affect healing of damaged tissues [52]. A better understanding of the mechanisms behind TLR2 regulation of immunity in infectious diseases could be a significant benefit for accelerating the discovery of TLR2-related vaccines or targeted therapeutic treatments. Therefore, we will review the function of TLR2 in infectious diseases by summarizing the mechanisms of TLR2 signaling and its regulation, describing TLR2-regulated host–mycobacterial interactions and discussing controversial results to suggest possibilities for future research and therapeutic applications.

## 2. Regulation of TLR2 Signaling 

The extracellular binding of the TLR LRR domain and its ligands stimulates the recruitment of adaptor proteins to interact with the intracellular TIR domain of TLRs to trigger the downstream signaling cascades. Several intracellular adaptor proteins are involved in relaying the signal from the cell membrane to the nucleus. The myeloid differentiation factor (MYD88) is a well-known adaptor protein that interacts with almost all TLRs, except TLR3 [3,53]. TIR domain-containing adaptor protein (TIRAP), which is also called MyD88 adaptor-like (MAL), is required in the activation of TLR2/1 or TLR2/6 signaling [54,55]. In addition to MYD88 and TIRAP, other adaptor proteins in mammalian cells include TIR domain-containing adaptor protein inducing interferon-β (TRIF) [56], TIR-containing adaptor molecule (TICAM) [57], TRIF-related adaptor molecule (TRAM) [57], and the sterile α- and armadillo motif-containing protein (SARM) [58]. The recruitment of distinct adaptor proteins can trigger different downstream signaling pathways. Several reviews have in detail discussed the known differences between downstream signaling pathways of the mammalian TLR receptors [3,59,60], and therefore we only briefly describe TLR2 signaling here and summarize it in Figure 3. After the interaction of TLR2 and its associated adaptor proteins, the IRAK complex is activated to recruit TRAF6 [61]. Activated TRAF6 triggers the activity of a complex of TAK1/ TABs to stimulate both the activation of the MAPKs and the IKK complex (IKK1, 2, and IKK-γ, also known as NEMO). The involved MAPKs families include JNKs and p38. The IKK complex promotes the nuclear translocation of NF-κB. In turn, this results in the production of pro-inflammatory cytokines by AP-1 and NF-κB, which controls inflammation and modulates cell survival and proliferation [39,62]. 

Accumulated evidence supports the notion that the activation of TLR2 signaling benefits the host defense against invading pathogens [63,64,65]. However, hyper-inflammation can be caused by excessive TLR signaling activation, which has been implicated in chronic inflammatory diseases, autoimmune diseases, and even aggravation of infectious diseases [66,67,68]. Hyper-inflammation is characterized by persisting leukocyte infiltration, which can be triggered by immoderate TLR2 signaling activation [69,70]. In our previous study, we found that fewer leucocytes were recruited to wounded tail fin tissue in both *tlr2* mutant and *myd88* mutant zebrafish larvae, which suggests *tlr2* and *myd88* are involved in responses to tail wounding [71]. Similarly, TLR2 deficiency in diabetic mice accelerates wound healing, which indicates that excessive activation of TLR2 signaling might be detrimental for wound healing [72]. Thus, it appears that TLR2 signaling needs to be tightly regulated by negative regulatory mechanisms that are still poorly understood. Some reviews have summarized many different mechanisms of negative regulation and their molecular components [10,73,74,75]. These negative regulators include ubiquitin ligases, deubiquitinases, transcriptional regulators, and microRNAs [75]. The mechanisms inhibiting TLR2 signaling are based on (1) the prevention of receptor–ligand binding; (2) the dissociation of adaptor complexes; (3) the inhibition of TLR2 downstream kinase signaling; and (4) the negative transcriptional regulation [73,74]. Soluble TLR2, which is a smaller isoform of the TLR2 protein, has been reported to be secreted by human monocytes, and can compete with TLR2 on cell membranes by binding its ligands, leading to the inhibition of signaling [76,77]. As a negative regulator that leads to dissociation of adaptor complexes, it has been shown that a short form of MyD88 (sMyD88) is unable to bind to IRAK4 and thereby its expression can inhibit NF-κB activation [78]. Another described mechanism for the inhibition of adaptor signaling is the induction of TIRAP degradation by the suppressor of cytokine signaling 1 (SOCS1) [79]. In terms of TLR2 downstream kinase signaling inhibition, the Toll-interacting protein (TOLLIP) inhibits the TLR2 signaling by targeting IRAK1 to suppress its phosphorylation or directly interacting with TLR2 [80]. Thus, TOLLIP is widely utilized as an inhibitor to inhibit TLR2 signaling [81]. IRAK-M is another IRAK inhibitor, which belongs to the IRAK kinase family, but cannot induce NF-κB activation [82]. In addition to the inhibitors targeting IRAKs, proteins binding to TRAF6 [83], namely A20 [84] (also called TNF- α induced protein 3, TNFAIP3), NLR family member X1 (NLRX1) [85,86], and the cylindromatosis protein (CYLD), have also been shown to be negative regulators of TLR2 signaling [87,88]. Furthermore, A20 and NLRX1 can also block the activation of the IKK complex [89]. The last category is the negative regulators of transcription. The transcription of some pro-inflammatory genes, such as IL-6, is negatively regulated by activating transcription factor 3 (ATF3) [90], by TLR-inducible IkB protein (IkBNS) [91], and B-cell CLL/lymphoma 3 (Bcl-3) [92]. It is not yet known how these negative regulators are controlled. It has been hypothesized that MYD88 might be involved in a feedback loop that are under control of TLR2 or other Toll-like receptors [93]. Therefore, it is very likely that TLR2 is an important control factor of negative regulation of transcription of genes involved in inflammation.

## 3. TLR2 Function in Immune Responses to Mycobacterial Infection 

### 3.1. Tuberculosis and Non-Tuberculosis Diseases Caused by Mycobacteria

Tuberculosis (TB) is a communicable disease, which is caused by infection with *Mycobacterium tuberculosis* (Mtb). In the 2021 WHO global TB report, it was reported that TB remains a major cause of ill health, which results in death, and currently its death toll is higher than that caused by other major infectious diseases except for COVID-19. In 2020, TB death tolls have increased due to a lack of TB diagnosis and treatment during the COVID-19 pandemic. Nontuberculous mycobacteria (NTM) diseases are defined as being caused by mycobacterial pathogens other than Mtb and *Mycobacterium leprae* [94]. NTM infectious diseases have recently attracted great attention because the disease prevalence has increased sharply since 2000 [95]. It is hard to combat TB and NTM infections due to the rapid increase in multi-drug resistant mycobacterial strains [96,97]. Therefore, there is an urgent need to discover novel preventive or therapeutic strategies for TB and NTM infectious diseases. Currently, host-directed therapies (HDTs) are one of the most promising strategies to combat NTM infectious diseases by making the NTM antibiotic treatment regimens more effective [96,98]. 

### 3.2. Which TLRs Play a Role in Defense against Mycobacteria?

TLRs affect host defense against invading mycobacteria by directly or indirectly participating in multiple biological processes, such as mycobacterial recognition, inflammatory responses, antimicrobial activity, and antigen presentation [99,100,101]. In mammals, TLR4 plays a role in defense against mycobacterial infection [38]. Immune cells can be activated by the interaction between TLR4 and Mtb components, such as secretory protein (Rv0335c), lipomannan (LM), HSPs 60, and 65 [65,102]. In a recent study, Thada et al. found that TLR4 can bind TLR8 to recognize TLR8 ligands produced by Mtb [103]. In chronic Mtb infection, it has been demonstrated that macrophage recruitment, pro-inflammatory responses, and the ability to eliminate mycobacteria were impaired in TLR4 deficient mice [104]. However, no pronounced susceptibility was found in TLR4 deficient mice in a high dose Mtb infection, compared to the wild-type controls [105]. Compared with TLR4, the function of TLR2 in mycobacterial infection is more complex, because TLR2 can recognize a large number of components on a mycobacterial membrane [99]. TLR2 forms a heterodimer with TLR1 or TLR6, which expands the range of its recognizable ligands [33,39]. In addition, TLR2 indirectly cooperates with the other TLRs to function in defense against mycobacterial infection. For example, TLR2/9 double knockout mice showed significantly enhanced susceptibility to Mtb infection when compared with TLR2^−/−^ mice and TLR9^−/−^ mice, which suggests that the cooperation between TLR2 and TLR9 adds to the resistances to Mtb infection [106]. TLR6 and TLR9 have also been demonstrated to play a role in resistance to *Mycobacterium avium* infection. The infection rates of *M. avium* are significantly higher in TLR6 or TLR9 deficient mice than in wild-type mice [107,108]. In a pathway analysis in a zebrafish larval mycobacterial infection model, the expression of the Tlr8 pathway connected to vitamin D signaling was strongly affected in a *tlr2* mutant, which implicates a link between Tlr2 and Tlr8 signaling [109]. TLR5, which is known to recognize flagellin from invading pathogens, might also be involved in mycobacterial infection [110]. This can be concluded from the observation that expression of *tlr5a* and *tlr5b* are both increased after *Mycobacterium marinum* or *M. avium* infection in zebrafish. More functions of TLR2 in innate immune responses to mycobacterial infection, will be discussed in detail below.

### 3.3. TLR2 Recognizes Mycobacterial Components 

As described in the introduction, TLR2 plays a crucial role in recognizing bacteria, such as Mtb, through their cell wall components [111]. TLR2 lipoprotein ligands of the cell surface of Mtb include 19-kDa lipoprotein (Rv3763, LpqH), 24-kDa lipoproteins (Rv1270c, LprA, and Rv1411c, LprG), and 38-kDa glycolipoprotein (PhoS1). Other categories of TLR2 ligands include lipoarabinomannan (LAM), lipomannans (LM), phosphatidylinositol dimannoside (PIMs) and trehalose dimycolate (TDM), and mycobacterial heat shock protein 70 (HSP70) [112,113]. The TLR2 ligands from Mtb are summarized and described in Table 1 [114]. As described in the introduction, these ligands activate macrophages by activating NF-κB through TLR2 (Figure 3). However, prolonged TLR2 signaling triggered by these ligands might help Mtb to evade immune surveillance. For example, long-term exposure of macrophages to LpqH, LprG, LprA, PhoS1, LM, and PIM leads to IL-10, IL-4, and TGF-β expression, which in turn inhibits the activation of macrophages [114]. Furthermore, it has been demonstrated that prolonged TLR2 signaling activated by LpqH and LprG inhibits the expression of MHC class II molecules and exogenous antigen processing for presentation to CD4^+^ T cells, which can be a basis for Mtb immune evasion (Figure 3) [115,116,117,118].

### 3.4. TLR2 Is Associated with the Susceptibility to Infection by Various Mycobacteria

The structural integrity of TLR2 is crucial for defense against invading pathogens. Single nucleotide polymorphisms (SNPs) in human TLR2 have been reported to associate with the increased susceptibility to infectious diseases [146]. For example, one of the TLR2 polymorphisms, Arg753Gln, has been demonstrated to lead to higher susceptibility to TB [147]. Moreover, the TLR2 polymorphism R753Q impairs the activation of TLR2 signaling upon *Mycobacterium smegmatis* infection [148]. These studies indicate that TLR2 has a function to protect against mycobacterial infectious diseases, although a small number of studies found no effect of other TLR2 polymorphisms [48,149,150]. Thus, animal models for TLR2 polymorphisms are required to investigate the phenotypic consequences of the polymorphisms in TB. 

Mice are widely used as animal models to study the function of TLR2 in resistance to mycobacterial infection. The evidence for the role of TLR2 in defense against NTM infection is still limited, and no correlation has been found between TLR2 polymorphisms and human susceptibility to NTM infection until recently [151]. Interestingly, TLR2^-/-^ mice were more susceptible to *M. avium* infection [152]. It has been demonstrated that TLR2-deficient mice, but not TLR4-deficient mice, were more susceptible to a high dose Mtb infection than wild-type mice [105,153,154]. It indicates that TLR2 and the other TLRs compensate for the lack of TLR4 function in mycobacterial infection. However, the results of the studies of the role of TLR2 in low-dose Mtb infection are controversial. As a result, it is not clear at which infectious stages TLR2 functions in defense against Mtb infection; it is undecided whether it functions at the acute infectious stage or chronic infectious stage. Some researchers hypothesize that TLR2 plays a protective role during Mtb chronic infection, while it does not affect Mtb acute infection [153,155,156]. In contrast, Tjärnlund et al. demonstrated that TLR2 has a function in Mtb acute infection (at 3 weeks post infection), but not in Mtb chronic infection (at 8 weeks post infection) [50]. Interestingly, other studies found no significant differences between TLR2-defective mice or wild-type mice upon low-dose Mtb infection, in both acute and chronic infection [105,106,157]. Several explanations can be proposed to account for these different findings: (1) different Mtb strains were used. Most researchers used the Mtb H37Rv strain, while some studies utilized the Mtb Kurono strain or the Mtb Erdman strain. (2) The application of different infection methods. Aerosol challenging is the most extensively used method [158], but some studies also use intranasal (i.n.), intravenous (i.v.), or intratracheal infection methods (i.t.). (3) Differences in the definition of acute or chronic infection. For example, how long is an infection considered a chronic infection? In some studies, 8 weeks was considered as chronic, while other studies considered 21 weeks as a threshold. In summary, the lack of standardization in mice studies has given rise to many uncertainties as to the function of TLR2 in defense against TB. Therefore, the other animal models require to be utilized to further study the TLR2 function in mycobacterial infection. To study the susceptibility of *tlr2* mutants to mycobacterial infection, we analyzed *tlr2* mutant zebrafish infected with *M. marinum* [109] and *M. avium* [159]. The results show that *tlr2* mutant zebrafish are more susceptible to mycobacterial infection. More information on the *tlr2* mutant zebrafish can be found in ZFIN (http://www.zfin.org).

## 4. TLR2 Function in Mediating the Host–Mycobacterial Interaction

### 4.1. Macrophage–Mycobacterial Interactions 

Macrophages are not only the primary cells to recognize the invasion of mycobacteria, but are also the main cellular components of granulomas [160]. TLR2 plays an essential role in mediating the interaction between macrophages and mycobacteria. At the early infection stage, TLR2 enhances the entrance of Mtb bacteria into macrophages by binding PE_PGR33, a mycobacterial protein from the Mtb [161]. The binding of TLR2 and PE_PGR33 can activate macrophages by inducing the expression of TNF-α and some other pro-inflammatory cytokines (Table 1), while it can also trigger the PI3K pathway that can impair the macrophage antimicrobial responses [161]. In addition to promoting inflammatory responses, TLR2 also plays a role in promoting apoptosis of macrophages [162], which is an important defense mechanism of the host against intracellular pathogens. For example, Sánchez et al. reported that the apoptosis triggered by Mtb infection depends on the TLR2 signaling pathway [163]. In addition, it has been demonstrated that the apoptosis induced by ESAT-6 is a TLR2-depentent event [164]. ESAT-6, an abundantly secreted protein of Mtb, is an important virulence factor. Furthermore, TLR2-dependent microRNA-155 (miR-155) expression is required to elicit macrophage apoptosis by *Mycobacterium bovis* Bacille Calmette and Guérin (BCG) [165]. The antimicrobial activity of macrophages is an essential function of the host to combat invading mycobacteria and is mediated by TLR2 [166]. In human macrophages, the stimulation of TLR2 by mycobacteria results in the upregulation of Cyp27B1 and VDR, which have a function in the induction of transcription of antimicrobial factors, such as the antimicrobial peptide cathelicidin [167,168]. In addition, TLR2 is also involved in inducing a late component of transcriptional response to *M. tuberculosis* [169]. Of note, mouse macrophages and human macrophages utilize different mechanisms to kill intracellular Mtb through TLR2 activation [51]. TLR2-mediated death of intracellular bacilli is an iNOS-dependent process in mouse macrophages, whereas in human macrophages it is iNOS-independent [50]. This underscores the need to develop alternative animal models, which are needed to confirm some of the TLR2-mediated mechanisms of triggering macrophage antimicrobial activity. There is only one in vitro study describing how mycobacteria can directly control macrophage migration by rearranging the cytoskeleton via activation of TLR2 [170]. In addition, Carlos et al. found that TLR2^-/-^ mice displayed increased bacterial burden, diminished myeloid cell recruitment, and defective granuloma formation [154]. This result can be linked to a study in *tlr2* mutant zebrafish that shows an increased susceptibility to infection by *M. marinum* [109]. In conclusion, TLR2 participates in mediating macrophage–mycobacteria interactions in many ways, including phagocytosis, apoptosis, antimicrobial activity, cell recruitment, and granuloma formation. However, the underlying mechanisms behind the TLR2-regulated processes are still not clear and need to be further studied. Except for macrophages, the activation of other hematopoietic and non-hematopoietic cells via TLR2 is also required for host resistance to mycobacterial infection [154,156]. By using various chimeric mice, Konowich et al. demonstrated that TLR2 signaling in hematopoietic cells plays a role in controlling bacterial burden and granuloma integrity, while TLR2 signaling in non-hematopoietic cells may play a role in promoting granulomatous inflammation and bacterial dissemination [156]. Interestingly, the adoptive transfer of TLR2 positive mast cells into these TLR2^-/-^ mice reversed the increased susceptibility of TLR2^-/-^ mice to Mtb infection [154].

### 4.2. Neutrophil–Mycobacterial Interactions 

In addition to macrophages, neutrophils are innate immune cells that have an important function in defense against mycobacterial infection. A large number of neutrophils can be detected in TB lesions and in the sputum of TB patients, which indicates that neutrophils play a crucial role during Mtb infection [171,172]. There is consensus that neutrophils are activated upon mycobacterial infection via TLR2-mediated recognition of LAM on the surface of bacteria (Table 1) [123]. However, the reports on the function of TLR2 in the regulation of the recruitment of neutrophils during mycobacterial infection are contradictory. In TLR2^-/-^ mice, the bacterial burden after Mtb infection was increased compared to the wild-type, and this was accompanied by an increased neutrophil influx in the lungs and tissue damage [173]. Conversely, after in vitro infection of alveolar epithelial cells by *M. bovis* BCG, the recruitment of neutrophils was significantly reduced by blocking TLR2 [174]. Moreover, injection of non-mannose-capped lipoarabinomannan (AraLAM), which is a TLR ligand from *M. smegmatis*, led to a stronger reduction of neutrophils influx in the pulmonary compartment in TLR2^−/−^ mice, compared to WT mice [145]. These results underscore the importance of studying the function of TLR2 in neutrophils migration in further detail.

## 5. Therapeutic Targeting of TLR2 Signaling in Diseases 

### 5.1. The Application of TLR2 Ligands in Mycobacterial Infectious Diseases

TLR2, as one of the most important representatives of PRRs, can recognize many mycobacterial PAMPs. Some of these TLR2 ligands constitute the main protein component of TB vaccines or adjuvants [161]. For example, the ESAT-6 and PPE18 proteins (Rv1196) are important components of the M72/AS01 and H56/IC31 vaccine candidates [161]. In addition, the mycobacterial MPT38 and PE_PGRS33 proteins have been reported to be TLR2-targeted secreted proteins that are promising pulmonary TB vaccines [161,175]. At present, *M. bovis* BCG remains the only available vaccine for TB, but it is only validated for prevention of TB in children [176,177]. Furthermore, there is no effective vaccine for preventing infectious diseases caused by NTM strains. 

The modulation of TLR2 signaling has become a popular approach for the design of host-directed therapeutics against NTM infectious diseases. In a recent study, the expression of a soluble CD157 protein (sCD157) correlates with the bactericidal activity of the human macrophages, which is a TLR2-dependent process [98,178]. The enhanced macrophage bactericidal activity by sCD157 stimulation was found to be based on the TLR2-dependent production of ROS [98,178]. This indicates that the TLR2 ligand, sCD157, might be an HDT to control mycobacterial infection [98,178]. A recent review described in detail how TLR2 could be used as a therapeutic target to cure bacterial infections [9]. TLR2 ligands from mycobacteria constitute a large group of natural TLR2 agonists and TLR2 antagonists (see Table 1). These TLR2 agonists or antagonists can be used to study the function of TLR2 in infectious diseases, and they also provide new possibilities as potential therapeutics that target TLR2 signaling to treat hyper-inflammation. For example, the recombinant PPE18 protein (rPPE18), which is a TLR2 ligand derived from Mtb, has been demonstrated to be a promising novel therapeutic to control sepsis [179], because rPPE18 significantly decreases the secretion of serum pro-inflammatory cytokines and reduces organ damage in mice infected with high doses of *E. coli* bacteria [179]. 

### 5.2. TLR2 as a Therapeutic Target in the Other Diseases

TLR2 is not only investigated as a promising therapeutic target for mycobacterial infectious diseases, but also for other diseases (see Figure 4). In recent studies, TLR2 was reported to be a potential therapeutic target for COVID-19 [180,181,182,183]. A spike (S) protein is a main structural protein of SARS-CoV-2, which induces the inflammatory responses that are dependent on TLR2 [180]. Since death and severe cases of COVID-19 are associated with a hyper-inflammatory response [184], TLR2 antagonists can be tested for their capacity to alleviate the hyper-inflammatory response in SARS-CoV-2 infection or S protein stimulation. In addition, TLR2 has been demonstrated to be involved in the regulation of immune responses in autoimmune inflammatory diseases [185,186], kidney inflammaging [187], tissue injuries [71,188], gut microbiome dysbiosis [93,189], diabetes [11] and cancers [190,191,192].

In some hyper-inflammatory diseases, such as autoimmune inflammatory disease, kidney inflammaging, and tissue injuries, TLR2 signaling is over-activated, which is detrimental for the prognosis of diseases and tissue repair. It has been shown that TLR2 functions in the modulation of intestinal serotonin transporter (SERT) function activated by enteropathogenic infections [189]. This indicates that regulating the intestinal serotonergic system can be therapeutically exploited to mitigate other enteropathogenic infections leading to Crohn’s disease [189]. In this case, also TLR2 inhibition by using its antagonists could be considered. 

TLR2 plays a dual role in immune responses to cancer cells. The activation of TLR2 has been demonstrated to be supportive in anti-cancer immunity [193]. However, excessive TLR2 activation can also lead to promoting cancer progression [193]. For TLR2 to be developed as a therapeutic target for cancers, its role in oncogenesis needs to be further elucidated. Essentially, the potential of using TLR2 as a therapeutic target and the application of its agonists or antagonists is dependent on whether combatting the particular disease is mostly benefitting from stimulating hyper-inflammatory or immunosuppressive responses.

## 6. Zebrafish as a Model to Investigate TLR2 as a Therapeutic Target 

### 6.1. General Advantages of the Zebrafish Larval Model

In recent decades, numerous disease models have been established using zebrafish larvae to study developmental processes, such as hematology [194,195], and pathogenic processes in cancer [196] and infectious diseases [197]. Zebrafish models have contributed to uncovering pathogenic mechanisms and to the discovery and efficacy screening of innovative drugs [198,199]. As an animal model, the zebrafish animal model possesses various advantages. The zebrafish larvae already have a functional innate immune system within 5 days post-fertilization, whereas the adaptive immune system is still not functional, providing a great advantage for studying the mechanisms of acute inflammation [200]. Moreover, its optical transparency and small size are the most significant advantages of the zebrafish embryos and larvae, because it provides an ideal in vivo system to directly observe cell–cell or cell–microbe interactions [200]. This is very difficult to achieve in other vertebrate models. In addition, the large number of zebrafish offspring make it trackable for omics studies of large groups of larvae. 

### 6.2. New Insights of the Tlr2 Function Learned from the Zebrafish Model

TLR2 polymorphisms increases the susceptibility to mycobacterial infection in the human population, although there is a small number of studies that found no effect of it [146,150]. In addition, there is still controversy about the role of TLR2 in host defense against Mtb in several rodent studies [50,153,155]. To this end, a *tlr2* knockout zebrafish to study Tlr2 function in innate immune defense during mycobacterial infection was developed. The transcriptome of homozygous mutant larvae and that of heterozygote larvae in the absence of infection was examined, and there were marked differences in the gene expression profiles of *tlr2^−/−^* zebrafish larvae and its control siblings. For example, pathway analysis showed that genes involved in glycolysis were particularly affected [109]. This result is consistent with a previous study in the human in vitro and mice in vivo models, which showed that TLR2 plays a key role to switch the host cellular metabolism toward aerobic glycolysis after Mtb infection [201]. In accordance, a previous study using zebrafish also suggested that MyD88 plays a role in metabolism [93]. This study also showed that Tlr2 and its adaptor MyD88 are crucial for the response of the host to the microbiome [93]. This indicates that the different gene expression profiles found in the mycobacterial infected *tlr2* mutant may part be caused by a dysfunctional response to the microbiome.

In a previous study, the role of Tlr2 in defense against *M. marinum* infection in zebrafish was investigated by injection of these bacteria in *tlr2* loss-of-function mutants and their homo- and heterozygote siblings. The bacterial burden was significantly higher in *tlr2* mutants and was accompanied with a higher extracellular bacterial burden and less granulomas than in *tlr2^+/−^* and wild-type larvae at 4 dpi [109]. This result is consistent with previous studies in mice that show a function of Tlr2 in zebrafish host defense [153,155,156]. In addition, transcriptome analysis showed that the number of up-regulated and down-regulated genes in response to infection was greatly diminished in infected *tlr2* mutant zebrafish compared to their heterozygous sibling controls. Moreover, many signaling pathways that have been demonstrated to be linked to TB in humans are differentially regulated in *tlr2* mutant zebrafish larvae. For example, the Tlr8 signaling pathway was strongly affected in infected *tlr2* mutant zebrafish, which indicates that Tlr2 signaling is connected to the function of Tlr8. In addition, the vitamin D receptor pathway genes were down-regulated in *tlr2* mutant zebrafish. It has been demonstrated that vitamin D plays an important role to control TB infection [202]. Therefore, the hyper-susceptibility of *tlr2* mutants to *M. marinum* infection could be caused by aberrant vitamin D signaling. Chemokines constitute the other gene category, which was affected by the *tlr2* mutation during *M. marinum* infection. In previous work, Torraca et al. demonstrated that the Cxcr3–Cxcl11 axis was involved in macrophage recruitment after *M. marinum* infection in zebrafish larvae [203,204]. In agreement, the expression levels of *cxcl11aa* and *cxcl11ac* were significantly lower in the *tlr2* mutants after infection. This result shows a clear connection between the Tlr2 function and macrophage chemotaxis. Moreover, it needs to be investigated whether the changes of leukocyte migration behavior in *tlr2* mutants are due to alterations of signals from the infection site or whether they are caused by cell-autonomous defects in the migratory abilities of the myeloid cells in the *tlr2* mutant. Cell transplantation techniques can be applied to investigate the non-intrinsic and intrinsic functions of myeloid cells in the *tlr2* zebrafish mutant after wounding and mycobacterial infection.

## 7. Future Perspectives

In addition to TB, NTM infectious diseases have recently attracted wide attention because their prevalence has increased in recent decades [95]. Although there are existing treatments for NTM infectious diseases, the treatment regimens are long and there is a high frequency of multi-drug resistant cases [96]. Therefore, there is an urgent need to develop models to discover novel prevention and therapeutic strategies for patients infected with NTM bacteria. Zebrafish is an ideal model for investigating the pathogenic mechanism of NTM infection and effectively screening new medicine for NTM diseases, because of its optical transparency and genetic tractability. Therefore, NTM infectious zebrafish models can be exploited in the future [96]. 

By using the zebrafish model, the mechanism of *tlr2* functions in host defense against a large diversity of mycobacteria can be further studied. For example, it can be investigated whether *tlr2* differentially regulates leukocyte migration behavior after infection by distinct mycobacterial species. To investigate cell migration behavior regulated by *tlr2*, real-time live imaging can be utilized to uncover novel phenotypes [71]. To determine the effect of *tlr2* mutation on mycobacterial infection, future investigations should focus on its role in controlling the function of chemokines in the different infection stages [159]. To confirm if the observation of cell migration behavior in *tlr2* mutant zebrafish are also translatable to the situation in human cells, in vitro cell tracking experiment should be conducted using primary and cultured myeloid cells in which TLR2 is inactivated. Moreover, to further explain the mechanistic basis of the differences in cell migratory behaviors, mathematical models can provide new insights. Chemokine and ROS gradients can be modeled by partial differential equations (PDEs). These can be incorporated into cell chemotaxis models, such as random walk models, phase field models, or the Cellular Potts model, with varying degrees of cell resolution, to study leukocyte migration. Such models could provide quantitative insights into how chemokines and ROS gradients affect the migration behavior of the leukocytes, and how the cells change these gradients by binding or secretion of chemokines or by absorption and metabolization ROS [205], which is known to affect the robustness of chemotaxis [206]. Using the Bayesian inference on tracking data, one can infer a number of chemotaxis parameters, such as the flow rate, diffusion coefficient, and production time of the chemoattractant [207]. 

Considering the broad function of TLR2 in many diseases (Figure 4), a further understanding of its function is of great importance. It is likely that TLR signaling integrates control of innate immune responses and metabolism, as suggested by various studies on rodents and zebrafish [109,208,209]. Therefore, theoretical modelling that integrates transcriptional and metabolic responses, for example, applied by van Steijn et al., will be useful to understand the effects of therapeutics that target TLR signaling [210]. Better physiology-based models will make it possible to investigate the effects of important factors, such as exercise and the circadian rhythm on the control function of TLR signaling, which, to date, have been understudied [211,212].

## Figures and Tables

**Figure 1 biology-11-00246-f001:**
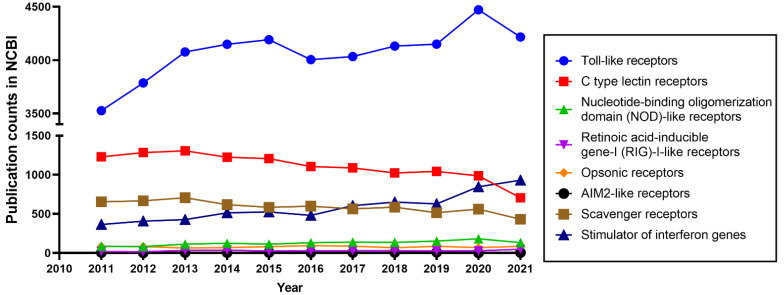
Publication counts for eight PRRs in the PubMed database (National Library of Medicine, National Center for Biotechnology Information, Bethesda, MD, USA). The full name of the eight PRRs was used as a keyword in PubMed. Access to Pubmed is through the link: https://pubmed.ncbi.nlm.nih.gov/. The results of the publication counts are sorted and exported by “Year”. The publication counts from the last 10 years (from 2011–2021) of 8 PRRs are shown in this Figure.

**Figure 2 biology-11-00246-f002:**
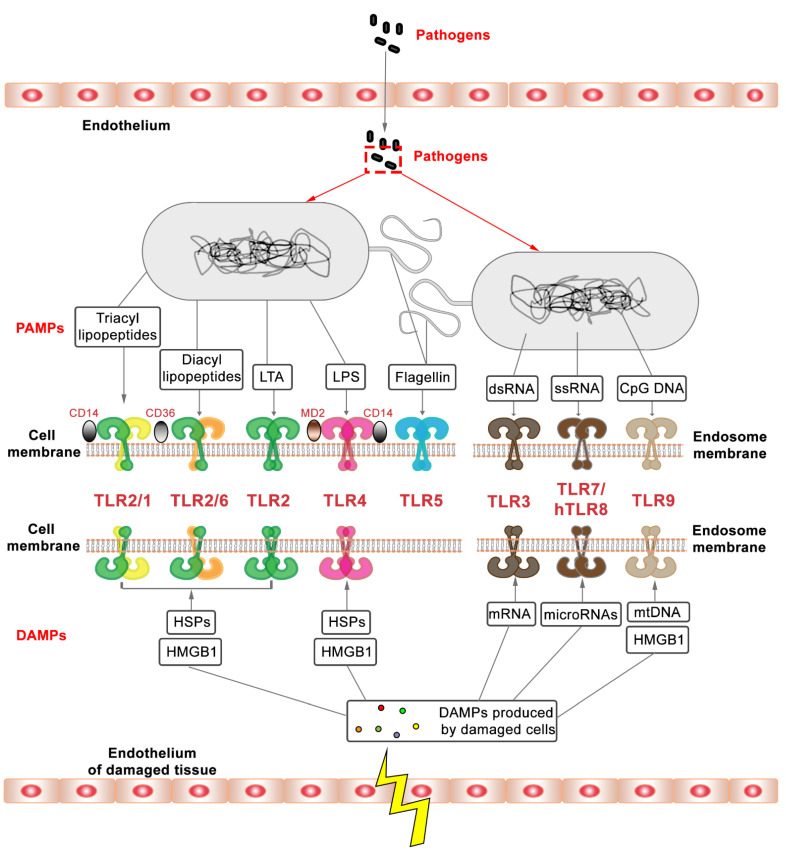
TLRs and their ligands. TLRs can recognize PAMPs from invading microbial pathogens and DAMPs from infected or damaged tissue. TLR2, its heterodimers, and TLR4 recognize pathogens through their cell wall surface components. TLR2 dimerizes with TLR1 or with TLR6 to sense triacyl or diacyl lipopeptides, and lipoteichoic acid (LTA) on the cell wall of Gram-positive bacteria and mycobacteria [37,38,39,40]. The process of the recognition of triacyl or diacyl lipopeptides by heterodimers requires the participation of accessory molecules. For example, CD14 and CD36 are well characterized as ligand delivery molecules that enhance TLR2 responses to ligands especially with a lower concentration of ligands, although the participation of these molecules is not essential [39,40]. TLR4 senses Gram-negative bacteria through the lipopolysaccharide (LPS) located on their outer membrane [38]. During this process, the formation of a complex of TLR4 with MD2 and CD14 is essential for recognizing LPS [38,41]. TLR5 functions in the recognition of flagellin from bacterial surfaces. There is still relatively little knowledge about the function of TLRs in the recognition of DAMPs compared with its function in the recognition of PAMPs. TLR3, 7, and 9 have been reported to play a role to sense nucleic acids released from damaged cells [42,43]. It has been demonstrated that TLR2 and TLR4 can be activated by the intracellular proteins or extracellular matrix components released from damaged cells [42,43]. It is controversial as to whether DAMPs directly interact with extracellular TLRs during this DAMPs recognition process [42]. Evidence suggests that recognition can be indirect, for instance, by the involvement of high-mobility group box 1 protein (HMGB1), which is a widely studied endogenous danger signal that induces inflammatory response through its interaction with DAMPs recognized by TLR2, TLR4, and TLR9 [44,45].

**Figure 3 biology-11-00246-f003:**
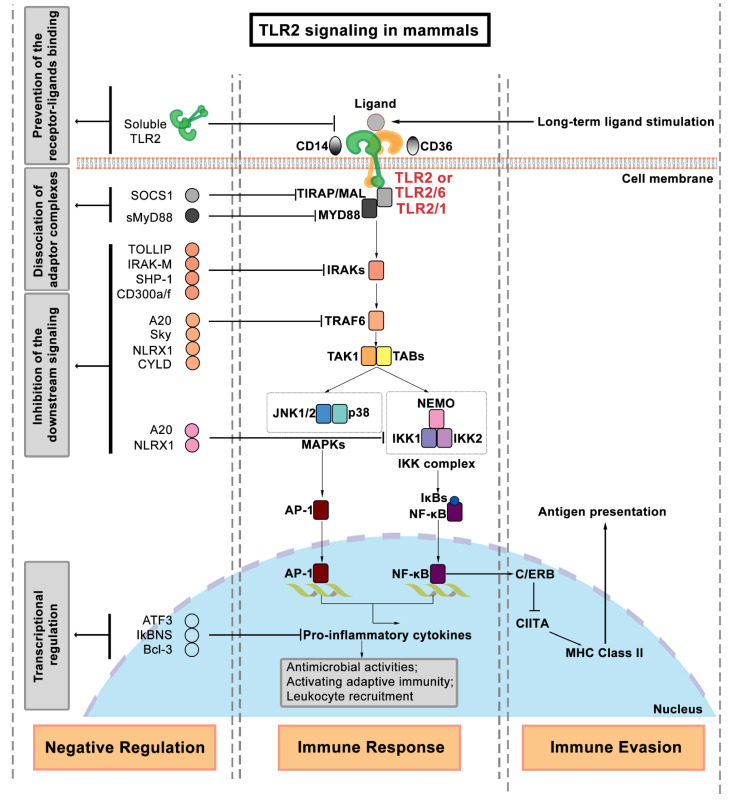
A brief overview of the TLR2 signaling pathway in mammals [39,62]. TLR2 or its heterodimers are located on the cell membranes. The TLR2 signaling activation through TLR2/1 requires the participant of accessory molecule CD14, while TLR2/6 requires CD36. The TLR2 signaling pathway is activated after TLR2 ligand recognition (PAMPs or DAMPs). Subsequently, the adaptor proteins, MYD88 and TIRAP/MAL, are recruited. After a series of cascades involving NF-κB and MAPKs, various transcription factors are activated to induce pro-inflammatory cytokines. Of note, the shown TLR2 signaling components are not exclusive for this TLR receptor, and the phosphorylation and ubiquitination processes are not mentioned in this Figure. The expansion of gene symbols and more gene information can be found in the HUGO Gene Nomenclature Committee website (HGNC, www.genenames.org) or the Mouse Genome Database (MGD, www.informatics.jax.org).

**Figure 4 biology-11-00246-f004:**
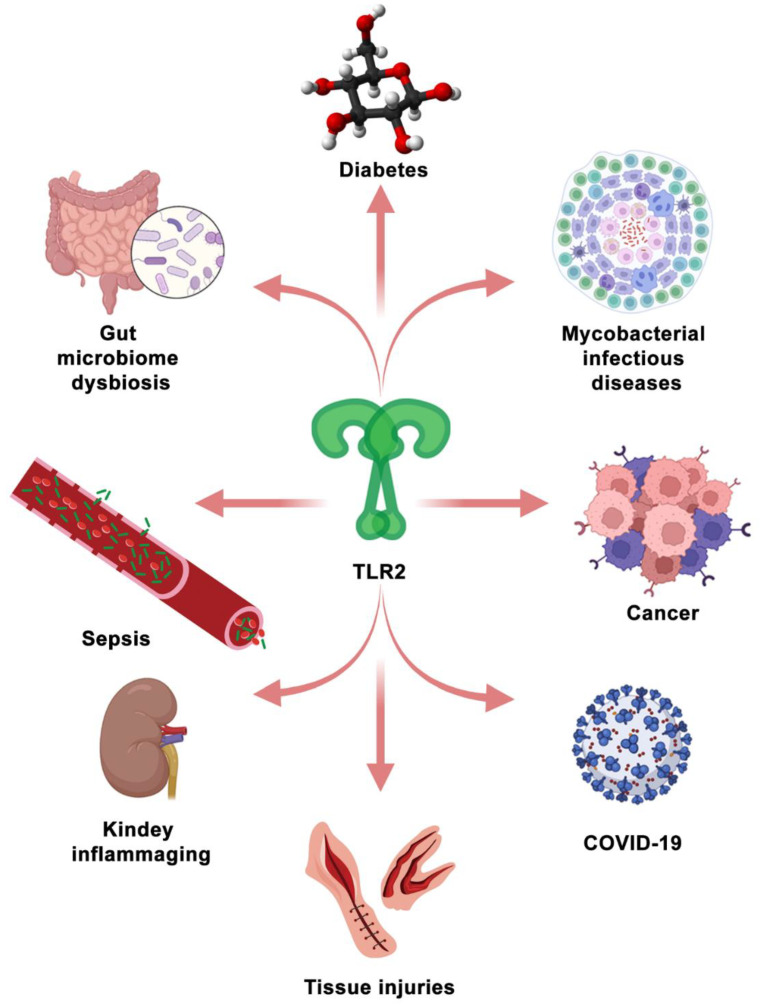
TLR2, which serves as a PRR, participates in the immunomodulation of some representative diseases. The Figure is made by BioRender.

**Table 1 biology-11-00246-t001:** Mycobacterial ligands of TLR2.

Spp	Ligand(s)	Abbreviation	PRRs	Accessory Molecules	Observations	Ref.
** *M. tuberculosis* **	** *Lipoproteins* **					
	19-kDa lipoprotein (Rv3763)	**LpqH**	TLR2/1	CD14	Inhibits MHC expression and antigen processing; IFN-γ-induced genes are inhibited by prolonged LpqH stimulation	[115,116,117]
	24-kDa lipoprotein (Rv1270c)	**LprA**	TLR2/1	CD14/CD36	Induces cytokine response and regulates antigen presenting cell functions	[117,119]
	24-kDa lipoprotein (Rv1411c)	**LprG**	TLR2/1; TLR2	CD14	Long-term exposure of LprG inhibits the processing of MHC-II antigen; Short-term exposure of LprG induces the production of TNF-α	[117,118]
	24-kDa lipoprotein (Rv1016c)	**LpqT**	TLR2	Unknown	Induces TLR2-dependent apoptosis in macrophages and inhibits MHC expression and antigen processing	[120]
	38-kDa glycolipoprotein	**PhoS1**	TLR2/1, TLR4	Unknown	Activates the ERK1/2 and p38 MAPK signaling, which in turn induces TNF-α and IL-6 expression	[117,121]
	Lipoylated and glycosylated Mtb lipoprotein (Rv2873)	**MPT83**	TLR2	unknown	MPT83-induced cytokine production is decreased in TLR2 defective mice	[122]
	** *Lipoglycans/Glycolipids* **					
	Lipoarabinomannan	**LAM**	TLR2/1; TLR2	CD14	Mtb LAM induces the production of pro- and anti-inflammatory cytokines to activate neutrophils	[111,123]
	Arabinosylated lipoarabinomannan	**AraLAM**	TLR2	Unknown	Induces pro-inflammatory responses	[124]
	Lipomannans	**LM**	TLR2/1; TLR2;	CD40/CD86	Induces TNF-α and NO secretion to activate macrophages	[125,126]
	phosphatidylinositol dimannoside	**PIM2/6**	TLR2	Unknown	Induces the expression of TNF-α to activate macrophages	[111,127]
	Trehalose dimycolate	**TDM**	TLR2	CD14/MARCO	Induces NF-κB signaling	[128]
	** *Others* **					
	Heat shock protein 70	**HSP70**	TLR2	Unknown	Inhibits the secretion of IL-6 in TLR2-deficient macrophages	[129]
	55-kDa flavin containing monooxygenase (Rv3083)	**MymA**	TLR2	CD40/CD80/ CD86/HLA-DR	Upregulates the expression of TLR2 and its co-simulatory molecules Activates macrophages by inducing TNF- α and IL-12	[130]
	PE_PGRS proteins (Rv1818c)	**PE_PGRS33**	TLR2	CD14	Contributes to Mtb entering macrophages by interacting with TLR2	[131,132]
	Secreted antigenic targets of 6-kDa (ESAT-6)family proteins (Rv1198)	**EsxL**	TLR2	Unknown	Induces TNF-α and IL-6 through TLR2-dependent NF-κB and MAPK signaling	[133]
	PE/PPE protein (Rv1196)	**PPE18**	TLR2	Unknown	Interacts with TLR2 to produce IL-10 and SOCS3 to in turn inhibit TLR2 signaling	[134,135]
	PE/PPE protein (Rv1789)	**PPE26**	TLR2	CD80/CD86	Activates macrophages by inducing pro-inflammatory cytokines TNF-α, IL-6, and IL-12	[136]
	PE/PPE protein (Rv1808)	**PPE32**	TLR2	Unknown	Induces both anti-inflammatory cytokine IL-10 and pro-inflammatory cytokines TNF-α and IL-6	[137]
	PE/PPE protein (Rv3425)	**PPE57**	TLR2	CD40/CD80/ CD86	Activates macrophages by inducing pro-inflammatory cytokines TNF-α, IL-6, and IL-12	[138]
	Leucine-responsive regulatory protein	**Lrp**	TLR2	Unknown	Inhibits LPS-induced pro-inflammatory cytokine IL-12 and TNF-α production	[139]
** *M. avium* **	Glycopeptidolipids	**GPLs**	TLR2, TLR4	Unknown	Promotes the activation of macrophages dependent on TLR2 and MYD88 TLR2 recognition of GPLs is dependent on specific acetylation and methylation patterns	[140,141,142]
** *M. abscessus* **	Glycopeptidolipids	**GPLs**	TLR2	Unknown	The switch of Mab from the smooth to the rough morphotype depends on the presence of bacterial surface GPLs	[143]
** *M. smegmatis* **	Phosphoinositol-capped LAM	**PILAM**	TLR2/ TLR1	Unknown	High affinity binding to TLR2 and strong pro-inflammatory responses	[125,144]
	Arabinosylated lipoarabinomannan	**AraLAM**	TLR2	CD14?	The lung inflammation induced by AraLAM is diminished in TLR2-deficient mice	[145]
	Dimannoside hosphatidyl-myo-inositol mannosides	**PIM2/6**	TLR2	Unknown	Induces the expression of TNF to activate primary macrophages	[127]

## Data Availability

Not applicable.

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
