# Peer review of "The Role of TLR2 in Infectious Diseases Caused by Mycobacteria: From Cell Biology to Therapeutic Target"

_biology, 2022, doi:10.3390/biology11020246_

Round 1

Reviewer 1 Report

The manuscript by Wanbin Hu et al. reviews the recent literature on the role of TLR2 in infectious diseases caused by mycobacte-2 ria.

The subject is very interesting and relevant. however, and even though the manuscript is well written, the authors are advised to seek the assistance of a native speaker and/or an experienced editor before the manuscript is accept for the pubblication. 

Author Response

Comments and Suggestions for Authors:

The manuscript by Wanbin Hu et al. reviews the recent literature on the role of TLR2 in infectious diseases caused by mycobacteria. The subject is very interesting and relevant. however, and even though the manuscript is well written, the authors are advised to seek the assistance of a native speaker and/or an experienced editor before the manuscript is accept for the publication. 

Response: We have asked a native speaker and an experienced academic paper writer for checking the manuscript.

Reviewer 2 Report

The Toll-like receptor 2 (TLR-2) is a very important component of the immune system and there is a lot of interest about this receptor as a target for developing immunotherapeutic and vaccine platforms. In this light, the current article has high importance. The authors here discussed the involvement of TLR-2 receptors in the disease states due to Mycobacterium Tuberculosis infections. The article first introduced different types of TLRs that are associated in the innate defense mechanism against any pathogens and then established the importance of TLR-2 in the current context with due information and literature references. The authors have described the involvement of TLR-2 in the Mycobacterium infection states in detail with ample examples. AT the end, the article has also discussed the scope of this target in future therapeutic designs. The article is well written with sufficient references and intriguing discussion that would draw the attention of the readers. The comprehensive discussion along with the compiled information will also be helpful to design new research around TLR-2. Hence, I recommend the current article to be published in its present form after complying with the following suggestions:

  1. If figure 1 is not created by the authors, please acquire a permission from NCBI for reproducing the figure in the current article.
  2. If figure 2 and 3 was published earlier, even by the same authors, please acquire a permission from the respective publishers to reproduce them in the current article.

Author Response

1.If figure 1 is not created by the authors, please acquire a permission from NCBI for reproducing the figure in the current article.

Response 1: The count of publications in NCBI is open source. Now we have mentioned how the data was collected in Fig 1 legend.  

2. If figure 2 and 3 was published earlier, even by the same authors, please acquire a permission from the respective publishers to reproduce them in the current article.

Response 2: Fig 2 and Fig 3 are original. They were not published before.

Reviewer 3 Report

In this manuscript, the two authors try to thoroughly dissect the role of TLRs, and specifically TLR2, in mycobacterial infections. It is a rather tough read, as it’s quite long, and the writing is not that good due to the small but numerous little things that are written in a way that they’re just not described right, or result in different interpretation, or missing interpunction, etc. I do apologize to the authors in advance that the further down the manuscript I got, the more frustrated I became due to the writing style, and this is reflected in my later comments. I would advise the authors to look at this manuscript with a fresh pair of eyes after the holidays and improve the general writing, as the content of the review is actually really good. After that, this manuscript can be considered for publication in Biology.

General comments:

  • Please be consistent in using shorthands such as Tb, and with capitalizing words/gene names where applicable; sometimes zebrafish proteins are capitalized, sometimes they are not, same for gene names; please be consistent and in accordance with standard rules for gene and protein writing styles.
  • Please be consistent in removing superfluous instances of Mycobacterium; once mentioned once it does not have to be repeated every time but can be shortened to M.
  • Please be consistent in super/subscripting where applicable (ie. CD4+, TLR2-/-).
  • The English in this review isn’t great; please have a native speaking colleague read and suggest changes where applicable in addition to the ones suggested below, as I must have missed a couple, and also stopped adding repeat instances of the same syntax/typographical/etc mistake at some point because my suggestions became rather numerous.

Line 12 – directions (plural)

Line 15 - … the susceptibility of hosts to infections by invading pathogens.

Line 19-20 - … in other diseases, such as chronic and acute inflammatory ….

Line 22 – Moreover, in some studies, the function of TLR2 was shown to be controversial, and therefore, its role in several diseases is still inconclusive.

Line 33 – remove comma after cells; furthermore, I don’t fully agree with reliance on membrane localized PRRs, as several well-known PRRs are cytosolic. Strongly suggest to remove “membrane-localized”

Line 37 – recommend to change “PRRs are comprised…” to “PRRs can be divided in the following 8 well-characterized groups”

Line 39 – remove “the” before nucleotide-binding for consistency

Line 42 – while I agree that TLRs are extremely important, this does not follow from the data compiled in figure 1. Please remove “important and”, because it is indeed the most widely studied PRR.

Line 45 – Please re-name the figure, there’s 8 types of PRRs in there. Furthermore, in the subscript, please elaborate (briefly) as to how the data was collected and a date on which this analysis was performed.

Line 49 – remove space after targets or add intended reference.

Line 51 – Please rephrase to “The investigation of TLRs intensified after their function…” as it is more correct.

Line 52-54 – this description is not correct for the TLRs located in endosomes. Please rephrase.

Line 55 – summary = summarize

Line 56 – remove “the” before specific.

Line 61 – add comma after fish

Line 71 – please also italize typhi and use a lowercase “t”.

Line 73 – remove “with” before RNA, and please use “Staphylococcus and Streptococcus spp.” if applicable.

Line 75 – remove “still”

Line 77 – add comma after “instance”

Line 78 – remove comma after “regulator”

Line 80 – “the” should be “The” at the start of a sentence; or maybe better, start the sentence with “For instance,”

Line 83 – “humans”

Line 84 – “testing”

Line 86 – “TLRs and their ligands”

Line 87 – add a comma after “heterodimers”

Line 89 – remove , after bacteria

Line 90 – swap needs for requires

Line 96 – remove these

Line 99 – remove comma after TLRs and add comma after indirect

Lone 105 – remove comma after functions

Line 106 – suggest to change to “… underlying mechanisms of recognition”

Line 108 – suggest to change “determines” to “underlines”, as distribution per se does not induce different functions, but rather supports it.

Line 131 - - suggest to change to “Some reviews have in detail discussed the known…”

Line 137 – suggest to change “named” to “known as”

Line 138 – suggest to change “In the end” to “In turn, this results in the production of pro-inflammatory cytokines by AP-1 and NF-kB…”

Line 142 – suggest to change to “Accumulated evidence that supports the notion that activation of…”

Line 144 – remove comma after signaling

Line 146 – infection = infectious

Line 146 – “For example, TLR2 deficiency in diabetic mice accelerates wound healing, which ..”

Line 149 – regulation = regulatory

Line 151 – suggest to change to “These negative regulators include ubiquitin ligases, deubiquitinases, …..”

Line 157 – I like this figure but the quality is quite poor, please enhance the pixel density.

Line 158 – change “To be noted” to “Of note”

Line 167 – add a comma after ligands

Line 168 – add a comma after complexes

Line 181 – remove “the” before negative regulators, there are probably more out there than currently known (or described in this review)

Line 193 – “give rise to” = result in

Line 192 – add a comma after first death

Line 195 – add comma after situation

Line 208 – change to “affect host defense”

Line 217 – macrophage – singular, remove the S

Line 221 “is more complex, because TLR2 can recognize a large number…” also, please add a space before ref 90

Line 227 – remove shorthand M. avium as it is common to not repeat the whole species name for bacteria

Line 231 – “TLR5, which is well known to….”

Line 238, 245 – remove “we”

Line 244 – remove “briefly” as it is quite a long and extensive table.

Line 245 – add a comma after “introduction”

Line 252 – please make it a superscript +, and it’s T cells, plural form, so add an S

Table 1 – The layout on this table is horrible. I would suggest to make the species direction vertical, shorten the word species to Spp, which should leave some extra room to reformat the table. Also shorten References to Ref. Try to not use justifying spacing because it looks horrible here, and also do not use the indent that makes text jump (for instance under observations first line, inhib-ited – the ited should start at the same height as the first part of the sentence on the previous line, same goes for TLR2 in the 3rd line). Please try to make this table more appealing because this looks very bad, and that makes readers skip over it. Also look at spelling, syntax and other relevant typography, because I’m not going to point out all the (little) mistakes in the observations column.

Line 272 onward – why are references now sometimes bold and sometimes not? Please format everything to be the same and compliant with journal style, as this is very annoying.

Line 280, 285 – polymorphisms, plural, add S

Line 281 – “Thus, animal models for TLR2 polymorphisms are required” as there are more than 1 polymorphism to study.

Line 282 – you have previously shorthanded tuberculosis to Tb, please use consistently throughout the manuscript.

Line 283 –“Mice are widely used as animal models to…”

Line 286, 361, 367 – superscript -/- and remove mutant; they are either knockout or mutant mice, can’t be both.

Line 292 – an = the

Line 295 – if you don’t use WPI later, then don’t shorten it and just write it out in full in the next sentence.

Line 307 – end of this paragraph – this seems very forced just to add some of your own data. I see the relevance, but the way it’s written now, makes it seem like it was done to add some extra citations. Please add a reason or subsequent statement what this data means in a grander scheme to round out and finish the paragraph.

Line 313 – primer = primary

Line 315 – of = between

Line 323 – “Mtb infection depends on the TLR2…”

Line 326 – add comma after Mtb

Line 334 – add a space after [159]. And change “to be noted” to “of note”

Line 345 – “mast” with a lowercase m

Line 348 – “linked with” = linked to

Line 363 – suggest to change to “Conversely, after in vitro infection of alveolar epithelial cells by Mycobacterium bovis…” to split up the 2 italic parts.

Line 377 – BCG is used previously but not defined there, please move this broader naming to the first site of mention.

Line 388 – remove “for”

Line 390 – remove space after TLR2-

Line 391 and onwards – “ antagonists can be used to study the function of TLR2 in infectious diseases, but they also provide new possibilities as potential therapeutics that target…”

Line 396 – don’t start a sentence with Because if it doesn’t end with a statement that is a result of what follows Because. Link this sentence with 395 via a comma and then it’s fine.

Line 388 – this paragraph is rather an amalgation of things; please consider splitting it up by introducing a full break after the section summing up things (408) and after Crohn’s disease (415) to split it up in ~4 parts.

Line 400 – “The Spike (S) protein is a main structural protein of SARS-CoV-2, which induces inflammatory responses that are dependent on TLR2” -> please note that this is the case in innate cells only, but not in antigen-specific T cells, which function just fine with just TCR triggering by the antigen.

Line 403 – add a space before ref 171

Line 407 – inflammaging? Inflammation?

Line 408 – please connect auto and immune with a hyphen (auto-immune)

Line 431 – possess should be plural in this context, or it should read “the zebrafish animal model possesses”

Line 436 – remove extra spaces after the hyphens

Line 440 – remove we

Line 440 – as described in previous what?

Line 474 – de-italicize during

Line 483 – change to “There is an urgent need to develop…”

Author Response

Response to Reviewer 3 Comments

Comments and Suggestions for Authors

In this manuscript, the two authors try to thoroughly dissect the role of TLRs, and specifically TLR2, in mycobacterial infections. It is a rather tough read, as it’s quite long, and the writing is not that good due to the small but numerous little things that are written in a way that they’re just not described right, or result in different interpretation, or missing interpunction, etc. I do apologize to the authors in advance that the further down the manuscript I got, the more frustrated I became due to the writing style, and this is reflected in my later comments. I would advise the authors to look at this manuscript with a fresh pair of eyes after the holidays and improve the general writing, as the content of the review is actually really good. After that, this manuscript can be considered for publication in Biology.

General comments:

  • Please be consistent in using shorthands such as Tb, and with capitalizing words/gene names where applicable; sometimes zebrafish proteins are capitalized, sometimes they are not, same for gene names; please be consistent and in accordance with standard rules for gene and protein writing styles.

Response:

All abbreviations have been checked and changed into a consistent format.

All nomenclatures have been checked in the manuscript. Formatting gene and protein names are based on the websites:

human: www.genenames.org

mouse: www.informatics.jax.org

zebrafish: www.zfin.org

  • Please be consistent in removing superfluous instances of Mycobacterium; once mentioned once it does not have to be repeated every time but can be shortened to M.

Response: “Mycobacterium” has been removed after the bacteria was mentioned.

Please be consistent in super/subscripting where applicable (ie. CD4+, TLR2-/-).

Response: Superscription and subscription have been checked and changed into a consistent format.

  • The English in this review isn’t great; please have a nativespeaking colleague read and suggest changes where applicable in addition to the ones suggested below, as I must have missed a couple, and also stopped adding repeat instances of the same syntax/typographical/etc mistake at some point because my suggestions became rather numerous.

Response: We have asked a native speaker and an experienced academic paper writer to critically read the manuscript.

Line 12 – directions (plural)

Response: It has been corrected. See line 14.

Line 15 - … the susceptibility of hosts to infections by invading pathogens.

Response: It has been corrected. See lines 17-18.

Line 19-20 - … in other diseases, such as chronic and acute inflammatory ….

Response: It has been corrected. See line 24.

Line 22 – Moreover, in some studies, the function of TLR2 was shown to be controversial, and therefore, its role in several diseases is still inconclusive.

Response: It has been corrected. See lines 25-27.

Line 33 – remove comma after cells; furthermore, I don’t fully agree with reliance on membrane localized PRRs, as several well-known PRRs are cytosolic. Strongly suggest to remove “membrane-localized”

Response: The comma after cells has been deleted. “membrane-localized” has been removed. See line 37.

Line 37 – recommend to change “PRRs are comprised…” to “PRRs can be divided in the following 8 well-characterized groups”

Response: This sentence has been rephrased according to this comment. See lines 41-42.

Line 39 – remove “the” before nucleotide-binding for consistency

Response: “the” before nucleotide-binding for consistency has been removed. See line 43.

Line 42 – while I agree that TLRs are extremely important, this does not follow from the data compiled in figure 1. Please remove “important and”, because it is indeed the most widely studied PRR.

Response: “important and” has been removed. See line 46.

Line 45 – Please re-name the figure, there’s 8 types of PRRs in there. Furthermore, in the subscript, please elaborate (briefly) as to how the data was collected and a date on which this analysis was performed.

Response: Fig 1 has been renamed. We have mentioned in detail how the data was collected from NCBI. See lines 49-52.

Line 49 – remove space after targets or add intended reference.

Response: Intended references have been cited. See reference 9-11 and see line 57.

Line 51 – Please rephrase to “The investigation of TLRs intensified after their function…” as it is more correct.

Response: The sentence has been rephrased according to this comment. See line 59.

Line 52-54 – this description is not correct for the TLRs located in endosomes. Please rephrase.

Response: The description in the structure of TLR has been changed. “N-terminal ectodomain” and “C- terminal domain” are applied. The descriptions are the same between cell-membrane TLRs and endosomal TLRs. On the basis of these descriptions, we further illustrated the compositions of each domain. See lines 67-72.
See references:
-Behzadi P, Garcia-Perdomo HA, Karpinski TM. Toll-Like Receptors: General Molecular and Structural Biology. J Immunol Res. 2021;2021:9914854. Epub 2021/07/02. doi: 10.1155/2021/9914854. PubMed PMID: 34195298; PubMed Central PMCID: PMCPMC8181103.
-Mielcarska MB, Bossowska-Nowicka M, Toka FN. Cell Surface Expression of Endosomal Toll-Like Receptors-A Necessity or a Superfluous Duplication? Front Immunol. 2020;11:620972. Epub 2021/02/19. doi: 10.3389/fimmu.2020.620972. PubMed PMID: 33597952; PubMed Central PMCID: PMCPMC7882679.

Line 55 – summary = summarize

Response: It has been corrected. See line 72.

Line 56 – remove “the” before specific.

Response: “the” has been removed before “specific”. See line 73.

Line 61 – add comma after fish

Response: “,” has been added after “In teleost fish”, see line 76

Line 71 – please also italize typhi and use a lowercase “t”.

Response: Typhi has been changed into “typhi”, see line 87

Line 73 – remove “with” before RNA, and please use “Staphylococcus and Streptococcus spp.” if applicable.

Response: “with” has been removed before “RNA” and “Staphylococcus and Streptococcus spp.” has been applied. See lines 89-90.

Line 75 – remove “still”

Response: “still” has been removed and the sentence has been rephrased. See lines 91-92.

Line 77 – add comma after “instance”

Response: “,” has been added after “For instance”, see line 93.

Line 78 – remove comma after “regulator”

Response: “,” has been removed after “regulator”, see line 94

Line 80 – “the” should be “The” at the start of a sentence; or maybe better, start the sentence with “For instance,”

Response: “For instance” has been added in front of the sentence, see line 96

Line 83 – “humans”

Response: “human” has been changed into “humans”, see line 99

Line 84 – “testing”

Response: “test” has been changed into “testing”, see line 100

Line 86 – “TLRs and their ligands”

Response: “TLRs and its ligands” has been replaced by “TLRs and their ligands”, see line 102

Line 87 – add a comma after “heterodimers”

Response: “,” has been added after “heterodimers”, see line 103

Line 89 – remove , after bacteria

Response: “,” has been removed after bacteria, see line 105

Line 90 – swap needs for requires

Response: “needs” has been replaced by “requires”, see line 106

Line 96 – remove these

Response: “these” has been removed, see line 113

Line 99 – remove comma after TLRs and add comma after indirect

Response: A “,” has been deleted after “TLRs” and a “,” has been added after “indirect”, see lines 115 and 116  

Lone 105 – remove comma after functions

Response: “,” has been removed after “functions”, see line 121

Line 106 – suggest to change to “… underlying mechanisms of recognition”

Response: “the underlying recognition mechanisms” has been changed into “the underlying mechanism of recognition”, see line 123.

Line 108 – suggest to change “determines” to “underlines”, as distribution per se does not induce different functions, but rather supports it.

Response: “determines” has been replaced by “underlines”, see line 125

Line 131 - - suggest to change to “Some reviews have in detail discussed the known…”

Response: “Some reviews have discussed in detail …” has been changed into “Several reviews have in detail discussed the known”, see lines 155-156.

Line 137 – suggest to change “named” to “known as”

Response: “named” has been changed into “known as”, see line 162

Line 138 – suggest to change “In the end” to “In turn, this results in the production of pro-inflammatory cytokines by AP-1 and NF-kB…”

Response: The sentence has been rephrased according to the comment. See lines 163-164.

Line 142 – suggest to change to “Accumulated evidence that supports the notion that activation of…”

Response: The sentence has been changed into “Accumulated evidence that supports the notion that the activation of TLR2 signaling benefits the host defense against invading pathogens”, see lines 167-168.

Line 144 – remove comma after signaling

Response: The sentence has been rephrased, see lines 170-172. “Excessive activation of TLR signaling, can lead to over-expression of pro-inflammatory cytokines” has been changed into “hyper-inflammation can be caused by excessive TLR signaling activation”.

Line 146 – infection = infectious

Response: “infection” has been changed into “infectious”, see line 172

Line 146 – “For example, TLR2 deficiency in diabetic mice accelerates wound healing, which ..”

Response: The sentence has been rephrased. See line 177.

Line 149 – regulation = regulatory

Response: “regulation” has been changed into “regulatory”, see line 180.

Line 151 – suggest to change to “These negative regulators include ubiquitin ligases, deubiquitinases, …..”

Response: The sentence has been changed. See lines 182-183.

Line 157 – I like this figure but the quality is quite poor, please enhance the pixel density.

Response: The figure 3 has been replaced by a higher resolution figure. See line 213

Line 158 – change “To be noted” to “Of note”

Response: “To be noted” has been changed into “Of note”, see line 220

Line 167 – add a comma after ligands

Response: “,” has been added after the word “ligands”, see line 190

Line 168 – add a comma after complexes

Response: a “,” has been added after the word “complexes”, see line 191.

Line 181 – remove “the” before negative regulators, there are probably more out there than currently known (or described in this review)

Response: “the” has been removed before negative, see line 204.

Line 193 – “give rise to” = result in

Response: This paragraph has been rephrased according to the latest 2021 WHO global TB report, please see lines 232-236

Line 192 – add a comma after first death

Response: This paragraph has been rephrased according to the latest 2021 WHO global TB report, please see lines 232-236

Line 195 – add comma after situation

Response: This paragraph has been rephrased according to the latest 2021 WHO global TB report, please see lines 232-236

Line 208 – change to “affect host defense”

Response: “TLRs affect the host’s defense” has been changed into “TLRs affect host defense”. See line 247.

Line 217 – macrophage – singular, remove the S

Response: “macrophages” has been changed into “macrophage”, see line 256

Line 221 “is more complex, because TLR2 can recognize a large number…” also, please add a space before ref 90

Response: The sentence has been changed according to this comment, and a space has been added before the reference, see line 260

Line 227 – remove shorthand M. avium as it is common to not repeat the whole species name for bacteria

Response: “(M.avium)” has been deleted, see line 269. All shorthand writing has been checked in the manuscript.

Line 231 – “TLR5, which is well known to….”

Response: “TLR5, that is well known…” has been changed into “TLR5,which is known to…”, see line 274

Line 238, 245 – remove “we”

Response: “we” has been deleted in line 280 and line 287

Line 244 – remove “briefly” as it is quite a long and extensive table.

Response: “briefly” has been removed, see line 286

Line 245 – add a comma after “introduction”

Response: a comma has been added after “introduction”, see line 287

Line 252 – please make it a superscript +, and it’s T cells, plural form, so add an S

Response: “CD4+ T cell” has been changed into “CD4+ T cells”, see line 294. All Sub/superscripts have been checked in the manuscript.

Table 1 – The layout on this table is horrible. I would suggest to make the species direction vertical, shorten the word species to Spp, which should leave some extra room to reformat the table. Also shorten References to Ref. Try to not use justifying spacing because it looks horrible here, and also do not use the indent that makes text jump (for instance under observations first line, inhib-ited – the ited should start at the same height as the first part of the sentence on the previous line, same goes for TLR2 in the 3rd line). Please try to make this table more appealing because this looks very bad, and that makes readers skip over it. Also look at spelling, syntax and other relevant typography, because I’m not going to point out all the (little) mistakes in the observations column.

Response: The format of table 1 has been modified according to the comment. The grammar of the sentences in the “observations” have been checked. See Table 1. See line 298

Line 272 onward – why are references now sometimes bold and sometimes not? Please format everything to be the same and compliant with journal style, as this is very annoying.

Response: The format of the references has been checked and modified in the same way.

Line 280, 285 – polymorphisms, plural, add S

Response: “s” has been added after “polymorphisms” in lines 319 and 324

Line 281 – “Thus, animal models for TLR2 polymorphisms are required” as there are more than 1 polymorphism to study.

Response: The sentence has been modified according to this comment, see lines 320-321.

Line 282 – you have previously shorthanded tuberculosis to Tb, please use consistently throughout the manuscript.

Response: “tuberculosis” has been changed into “TB” except for the first time when it appears.

Line 283 –“Mice are widely used as animal models to…”

Response: The sentence has been modified, see line 322

Line 286, 361, 367 – superscript -/- and remove mutant; they are either knockout or mutant mice, can’t be both.

Response: “-/-” has been superscripted and “mutant” has been removed if there is already a “-/-”. “-/-” has been checked throughout the manuscript

Line 292 – an = the

Response: “an” has been changed into “the”, see line 332

Line 295 – if you don’t use WPI later, then don’t shorten it and just write it out in full in the next sentence.

Response: “WPI” has been deleted. See lines 336-337.

Line 307 – end of this paragraph – this seems very forced just to add some of your own data. I see the relevance, but the way it’s written now, makes it seem like it was done to add some extra citations. Please add a reason or subsequent statement what this data means in a grander scheme to round out and finish the paragraph.

Response: We have now given a reason why we would like to further study the function of TLR2 in zebrafish model, see lines 349-350

Line 313 – primer = primary

Response: “primer” has been changed into “primary”, see line 356

Line 315 – of = between

Response: “of” has been changed into “between”, see line 358

Line 323 – “Mtb infection depends on the TLR2…”

Response: “Mtb infection is depending on the TLR2 signaling pathway” has been changed into “Mtb infection depends on the TLR2”, see lines 366-367

Line 326 – add comma after Mtb

Response: a “,” has been added after “Mtb”, see line 369

Line 334 – add a space after [159]. And change “to be noted” to “of note”

Response: a space has been added after the reference, and “to be noted” has been changed into “of note”, see line 377

Line 345 – “mast” with a lowercase m

Response: “Mast” has been changed into “mast”, see line 408

Line 348 – “linked with” = linked to

Response: “linked with” has been changed into “linked to”, see line 395

Line 363 – suggest to change to “Conversely, after in vitro infection of alveolar epithelial cells by Mycobacterium bovis…” to split up the 2 italic parts.

Response: The sentence has been modified, see lines 420-421

Line 377 – BCG is used previously but not defined there, please move this broader naming to the first site of mention.

Response: The broader name has been moved at the first place where the name appears, see lines 371, 421, 436.

Line 388 – remove “for”

Response: “for” has been removed, see line 447.

Line 390 – remove space after TLR2-

Response: spaces have been removed after “TLR2-”, see line 450

Line 391 and onwards – “ antagonists can be used to study the function of TLR2 in infectious diseases, but they also provide new possibilities as potential therapeutics that target…”

Response: the sentence has been rephrased based on the suggestion, see lines 451-452

Line 396 – don’t start a sentence with Because if it doesn’t end with a statement that is a result of what follows Because. Link this sentence with 395 via a comma and then it’s fine.

Response: Two sentences have been combined in one sentence based on the comment, see line 455.

Line 388 – this paragraph is rather an amalgation of things; please consider splitting it up by introducing a full break after the section summing up things (408) and after Crohn’s disease (415) to split it up in ~4 parts.

Response: This section has been separated based on the comment. To make this section more clear, two subheadings have been added. See lines 428-486

Line 400 – “The Spike (S) protein is a main structural protein of SARS-CoV-2, which induces inflammatory responses that are dependent on TLR2” -> please note that this is the case in innate cells only, but not in antigen-specific T cells, which function just fine with just TCR triggering by the antigen.

Response:
Here we cited more references to support the point that TLR2 can be a potential therapeutic target for COVID-19, see line 463.
In the reference: Khan S, Shafiei M, Longoria C, Schoggins JW, Savani R, Zaki H. SARS-CoV-2 spike protein induces inflammation via TLR2-dependent activation of the NF-kB pathway. Elife. 2021;10. Epub 2021/12/07. doi: 10.7554/eLife.68563. PubMed PMID: 34866574.

Their biochemical studies revealed that S protein triggers inflammation via activation of the NF-κB pathway in a MyD88-dependent manner. Further, such an activation of the NF-κB pathway was abrogated in Tlr2-deficient macrophages. Consistently, administration of S protein-induced IL-6, TNF-α, and IL-1β in wild-type, but not Tlr2-deficient mice.

Line 403 – add a space before ref 171

Response: a space has been added in front of the reference, see line 466

Line 407 – inflammaging? Inflammation?

Response:

See the reference: Sepe V, Libetta C, Gregorini M, Rampino T. The innate immune system in human kidney inflammaging. J Nephrol. 2021. Epub 2021/11/27. doi: 10.1007/s40620-021-01153-4. PubMed PMID: 34826123; PubMed Central PMCID: PMCPMC8617550.

Inflammaging is an age-related long-term result of premature immune system senescence during a persistent, low-grade, non-resolving inflammatory state.

Line 408 – please connect auto and immune with a hyphen (auto-immune)

Response: “auto” and “ïmmune” have been connected, see line 468

Line 431 – possess should be plural in this context, or it should read “the zebrafish animal model possesses”

Response: “zebrafish possesses” has been changed into “the zebrafish animal model possesses”, see line 497.

Line 436 – remove extra spaces after the hyphens

Response: Extra spaces have been deleted after the hyphens, see line 502.

Line 440 – remove we

Response: “as we described in previous” has been removed, see line 507

Line 440 – as described in previous what?

Response: “as we described in previous” has been removed, see line 507

Line 474 – de-italicize during

Response: “during” has been de-italicized, see line 546

Line 483 – change to “There is an urgent need to develop…”

Response: “it” has been changed into “There”, see line 561

Reviewer 4 Report

The manuscript by Wanbin Hu, and Herman P. Spainkv discusses a very well known fact regarding host cells depending on pattern recognition receptors (PRRs) to recognize invading pathogens and initiate protective innate immune responses. The authors elaborate the role of TLR2 in mycobacterial infections. They further emphasizes that TLR2 perhaps have a dual role in both activation and suppression of innate immune responses. Several review articles have been written on role of TLR2 in infectious diseases as well as in mycobacterial infections. 

I have following concerns related to the article:

  1. Line 61- some TLR receptor have diversified—correct the language
  2. Line 55:we summary- it should be “summarize”
  3. Line 75-could be rewritten as crystal structure of TLR-Ligand complex
  4. Line 80-diversity “[24-26]. the structure of TLR2 in..”” fullstop and capital letter must be followed
  5. Line 111- what type of controversy?
  6. At some point it must be mentioned why only TLR2 signalling is chosen for discussion.
  7. “Accumulating evidence has been reported that the activation of TLR2 signaling is a...” Language must be reframed
  8. However, excessive activation of TLR signaling, can lead to over-expression ......its a repeated sentence, already mentioned previously.
  9. Line 143“However, excessive activation of TLR signalling....” here it may be elaborated what factors may contribute towards excessive, low or optimal TLR signalling.
  10. Line 151 “Negative regulators are including the....” language should be rewritten
  11. Line 155 “negative transcriptional regulation by 155 TLR2....”. Does TLR2 regulate itself negatively?
  12. Line 141 “2.2 Negative regulation in TLR2 signaling” What message does this subtitle intend to deliver?
  13. Line 193-194 “It has been reported that in recent years, incidence and death of TB are falling......” This sentence doesn’t make any sense
  14. Line 196-197 “diagnostic and treatment capacity” capacity may be replaced with some appropriate word
  15. Line 211 “TLR4 plays a role in defence against mycobacterial infection” Then why TLR2 signalling is only discussed and not TLR4 signalling? When the review article is about mycobacterial disease.
  16. At many instances the review lacks justifications and explanations. Some sentences are synthesized and left without proper discussions for example Line 225 -“it has been demonstrated that TLR9 can cooperate with 225 TLR2 to resist Mtb infection”. How does it co-operate?  Under what circumstances co-operation occurs and in what instances the co-operation is avoided? and so on.
  17. Line 227-228 “The infection rates of M. 227 avium are significant higher in TLR6 or TLR9 deficient.....” significant or significantly?
  18. Line 231 “TLR5, that is is well”... delete repeated word.
  19. Line 233 “tlr5a 233 and tlr5b a..” Why suddenly TLR is expressed in italics? What does a and b implicate?
  20. Line 287-288 “It has been demonstrated that TLR2 deficient 287 mice, but not TLR4 deficient mice, were more susceptible to a high dose Mtb infection 288 than wild type mice....” At this point it would be interesting to address the issue of pleiotropy and redundancy. As it has also been described that TLR deficient mice doesnot exhibit a clear phenotype because its function is compensated by other TLRs.
  21. Line 279 and 293 “that TLR2 plays a protective role during Mtb chronic infection..” is repeated several times with same words.
  22. Line 300-301 “The application of different infection methods”. This line does not make any sense?
  23. Page numbers are incorrectly allocated in the manuscript for example page 2 of 29, page 3 of 29 is repeated.( After table)
  24. Line 307-309 “susceptibility of tlr2 307 mutants to mycobacterial infection, we analyzed tlr2 mutant zebrafish....” Why again TLR is expressed in italics.
  25. Line 323 “y Mtb infection is de- 323 pending on the TLR2 ....” sentence and English language should be reframed.
  26. Line 348- why TLR is expressed in italics?
  27. Line 352 “underlying these function” reframe the sentence.
  28. Line 371” TLR2, as one of the most important representatives of PPRs....” What is PPR?
  29. Line 379 “vaccine for infection disease...” reframe the sentence
  30. Line 381 “TLR2 plays a dual role to trigger both pro-inflammatory and anti-inflammatory responses after infection..” How anti-inflammatory? Please give references
  31. Line 417 “However, extensive TLR2 signaling activation can also lead to pr.....” how can extensive TLR2 signalling occur?
  32. Line 439: “New insights of Tlr2 function learned from the zebrafish model” Why TLR2 is expressed in several different ways TLR2, Tlr2, tlr2 ...?. It seems as if the language is cut copied and pasted from the research article and submitted in a rush without proper re-synthesis of statements.
  33. Line 474: At some place chemokine is expressed as e Cxcl and at other place cxcl
  34. Line 483 “cases [87]. it is..” where is the line starting and stopping?
  35. Line 497: “whether they are caused cell-autonomous defects” line doesn’t make sense
  36. Recent references must be cited.
  37. All references number in text are not written with same font. (reference no 150-200 are bold letters)
  38. Main heading of figures is bold somewhere but un bold in some other figures. Homogeneity must be maintained.

Author Response

Response to Reviewer 4 Comments

Comments and Suggestions for Authors

The manuscript by Wanbin Hu, and Herman P. Spainkv discusses a very well known fact regarding host cells depending on pattern recognition receptors (PRRs) to recognize invading pathogens and initiate protective innate immune responses. The authors elaborate the role of TLR2 in mycobacterial infections. They further emphasizes that TLR2 perhaps have a dual role in both activation and suppression of innate immune responses. Several review articles have been written on role of TLR2 in infectious diseases as well as in mycobacterial infections. 

I have following concerns related to the article:

  1. Line 61- some TLR receptor have diversified—correct the language

Response: The sentence has been corrected, see line 77

2. Line 55:we summary- it should be “summarize”

Response: “summary” has been corrected with “summarize”, see line 72

3. Line 75-could be rewritten as crystal structure of TLR-Ligand complex

Response: “The crystal structure of TLRSs in complex with their ligands” has been changed into “Crystal structures of TLR-ligand complexes have been widely studied.” See line 91-92

4. Line 80-diversity “[24-26]. the structure of TLR2 in..”” fullstop and capital letter must be followed

Response: “For instance” has been added in the front of the sentence, see line 96

5. Line 111- what type of controversy?

Response: After the reference [48], we have given an example of a controversial result, see lines 130-134

6. At some point it must be mentioned why only TLR2 signalling is chosen for discussion.

Response: The reasons why we only focused on TLR2 signaling have been mentioned, see lines 134-140. 

7.“Accumulating evidence has been reported that the activation of TLR2 signaling is a...” Language must be reframed

Response: The sentence has been rephrased into “Accumulated evidence that supports the notion that the activation of TLR2 signaling benefits the host defense against invading pathogens”, see lines: 167-168.

8. However, excessive activation of TLR signaling, can lead to over-expression ......its a repeated sentence, already mentioned previously.

Response: The sentence has been rephrased into “However, hyper-inflammation can be caused by excessive TLR signaling activation”, see line 170

9. Line 143“However, excessive activation of TLR signalling....” here it may be elaborated what factors may contribute towards excessive, low or optimal TLR signalling.

Response: We have extended the explanations in the manuscript, see lines 172-176.

10. Line 151 “Negative regulators are including the....” language should be rewritten

Response: The sentence has been modified, see lines 182-183

11. Line 155 “negative transcriptional regulation by 155 TLR2....”. Does TLR2 regulate itself negatively?

Response: “by TLR2 and other factors as described below” has been deleted, see line 187

12. Line 141 “2.2 Negative regulation in TLR2 signaling” What message does this subtitle intend to deliver?

Response: Considering “negative regulation” also belongs to “regulation of TLR2 signaling”, the subtitles in this section have been deleted, see lines 143 and 166

13. Line 193-194 “It has been reported that in recent years, incidence and death of TB are falling......” This sentence doesn’t make any sense

Response: Indeed, the coronavirus disease (COVID-19) pandemic has reversed gains and set back the fight against TB by several years. We have modified the description of the death of TB based on the latest 2021 WHO global TB report, see lines 232-236

14. Line 196-197 “diagnostic and treatment capacity” capacity may be replaced with some appropriate word

Response: The sentence has been rephrased, see line 235

15. Line 211 “TLR4 plays a role in defence against mycobacterial infection” Then why TLR2 signalling is only discussed and not TLR4 signalling? When the review article is about mycobacterial disease.

Response: See lines 251-259, we already discussed the TLR4 function in mycobacterial infection. We explained the reasons afterward why we only focused on TLR2 signaling, see lines 259-264

16. At many instances the review lacks justifications and explanations. Some sentences are synthesized and left without proper discussions for example Line 225 -“it has been demonstrated that TLR9 can cooperate with 225 TLR2 to resist Mtb infection”. How does it co-operate?  Under what circumstances co-operation occurs and in what instances the co-operation is avoided? and so on.

Response: We have further discussed the cooperation between TLR2 and TLR9, see lines 264-267. Furthermore, we have checked through the manuscript to supplement proper reasons for some points.

17. Line 227-228 “The infection rates of M. 227 avium are significant higher in TLR6 or TLR9 deficient.....” significant or significantly?

Response: “significant higher” has been changed into “significantly higher”, see line 270

18. Line 231 “TLR5, that is is well”... delete repeated word.

Response: Extra “is” has been deleted, see line 274

19. Line 233 “tlr5a 233 and tlr5b a..” Why suddenly TLR is expressed in italics? What does a and b implicate?

Response: Based on the rule of nomenclature writing of zebrafish, full gene names are italicized (e.g., brass). Gene symbols are also italicized, with all letters in lower-case (e.g., brs). Protein symbols are not italicized, and the first letter is upper-case (e.g., Brs). See the website: https://www.biosciencewriters.com/guidelines-for-formatting-gene-and-protein-names.aspx

In zebrafish, the Tlr5 possesses two orthologs-Tlr5a and Tlr5b, which are both involved in the recognition of flagellin, see our previous publication: Common and specific downstream signaling targets controlled by Tlr2 and Tlr5 innate immune signaling in zebrafish.

20. Line 287-288 “It has been demonstrated that TLR2 deficient 287 mice, but not TLR4 deficient mice, were more susceptible to a high dose Mtb infection 288 than wild type mice....” At this point it would be interesting to address the issue of pleiotropy and redundancy. As it has also been described that TLR deficient mice does not exhibit a clear phenotype because its function is compensated by other TLRs.

Response: This point has been extended discussed in the manuscript based on the comment. See line 328-329

21. Line 279 and 293 “that TLR2 plays a protective role during Mtb chronic infection..” is repeated several times with same words.

Response: The sentence has been rephrased, see lines 317-318

22. Line 300-301 “The application of different infection methods”. This line does not make any sense?

Response: The sensitivity of different tissues in response to bacterial invasion is different.

See reference: Cooper AM. Mouse model of tuberculosis. Cold Spring Harbor perspectives in medicine. 2015 Feb 1;5(2):a018556.

In recent studies with the mouse model, one aspect of early infection was highlighted when it became clear that the acquired immune response largely ignores the arrival of the bacterium in the lung following aerosol infection.

Therefore, different infectious methods might provide different outcomes.

23. Page numbers are incorrectly allocated in the manuscript for example page 2 of 29, page 3 of 29 is repeated.( After table)

Response: The page numbers have been corrected.

24. Line 307-309 “susceptibility of tlr2 307 mutants to mycobacterial infection, we analyzed tlr2 mutant zebrafish....” Why again TLR is expressed in italics.

Response: Based on the rule of nomenclature writing of zebrafish, full gene names are italicized (e.g., brass). Gene symbols are also italicized, with all letters in lower-case (e.g., brs). Protein symbols are not italicized, and the first letter is upper-case (e.g., Brs). See the website: https://www.biosciencewriters.com/guidelines-for-formatting-gene-and-protein-names.aspx

25. Line 323 “y Mtb infection is de- 323 pending on the TLR2 ....” sentence and English language should be reframed.

Response: The sentence has been changed into “…by Mtb infection depends on the TLR2 signaling pathway”, see lines 366-367

26. Line 348- why TLR is expressed in italics?

Response: Based on the rule of nomenclature writing of zebrafish, full gene names are italicized (e.g., brass). Gene symbols are also italicized, with all letters in lower-case (e.g., brs). Protein symbols are not italicized, and the first letter is upper-case (e.g., Brs). See the website: https://www.biosciencewriters.com/guidelines-for-formatting-gene-and-protein-names.aspx

27. Line 352 “underlying these function” reframe the sentence.

Response: the sentence has been rephrased into “However, the underlying mechanisms behind the TLR2 regulated processes are still not clear and need to be further studied.” See lines 399-400

28. Line 371” TLR2, as one of the most important representatives of PPRs....” What is PPR?

Response: “PPRs” has been corrected by “PRRs”. See line 430

29. Line 379 “vaccine for infection disease...” reframe the sentence

Response: “vaccine for infection disease” has been changed into “vaccine for preventing infectious diseases..”, see lines 438-439

30. Line 381 “TLR2 plays a dual role to trigger both pro-inflammatory and anti-inflammatory responses after infection..” How anti-inflammatory? Please give references

Response: This sentence has been deleted due to it was mentioned several times. More references have been cited at the first place where this sentence appears, see lines 130-131

31. Line 417 “However, extensive TLR2 signaling activation can also lead to pr.....” how can extensive TLR2 signalling occur?

Response: “extensive” has been changed into “excessive”, see line 480

32. Line 439: “New insights of Tlr2 function learned from the zebrafish model” Why TLR2 is expressed in several different ways TLR2, Tlr2, tlr2 ...?. It seems as if the language is cut copied and pasted from the research article and submitted in a rush without proper re-synthesis of statements.

Response: TLR2, Tlr2 and tlr2 represent tlr2 in different species or gene/protein formats. Here are the rules of nomenclature writing:

Humans, gene symbols contain three to six italicized characters that are all in upper-case (e.g., AFP). Gene symbols may be a combination of letters and Arabic numerals (e.g., 1, 2, 3), but should always begin with a letter; they generally do not contain Roman numerals (e.g., I, II, III), Greek letters (e.g., α, β, γ), or punctuation. Protein symbols are identical to their corresponding gene symbols except that they are not italicized (e.g., AFP).

Mice and rats: Gene symbols are italicized, with only the first letter in upper-case (e.g., Gfap). Protein symbols are not italicized, and all letters are in upper-case (e.g., GFAP).

Fish: In contrast to the general rule, full gene names are italicized (e.g., brass). Gene symbols are also italicized, with all letters in lower-case (e.g., brs). Protein symbols are not italicized, and the first letter is upper-case (e.g., Brs).

See the website: https://www.biosciencewriters.com/guidelines-for-formatting-gene-and-protein-names.aspx

33. Line 474: At some place chemokine is expressed as e Cxcl and at other place cxcl

Response: Format of “cxcl” has been checked and modified based on the rule of nomenclature writing rules in zebrafish.

34. Line 483 “cases [87]. it is..” where is the line starting and stopping?

Response: “it is” has been changed into “There is”, see line 561

35. Line 497: “whether they are caused cell-autonomous defects” line doesn’t make sense

Response: We have moved this part into section 6, see lines 550-556

36. Recent references must be cited.
Response: All the references have been checked through the manuscript, and older references have been replaced by relative recent references.

37. All references number in text are not written with same font. (reference no 150-200 are bold letters)
Response: The format of the references has been checked and modified in the same way.

38. Main heading of figures is bold somewhere but un bold in some other figures. Homogeneity must be maintained.

Response: The format of the main heading of figures has been checked and modified in the same way through the manuscript

Round 2

Reviewer 3 Report

I would like to thank the authors for updating the manuscript and for acknowledging the peer review process. During the revisions, some extra commas and typos were added, but these can be fixed during typesetting. I think this manuscript is now suitable for publication.